# Idebenone Mitigates Traumatic-Brain-Injury-Triggered Gene Expression Changes to Ephrin-A and Dopamine Signaling Pathways While Increasing Microglial Genes

**DOI:** 10.3390/cells14110824

**Published:** 2025-06-01

**Authors:** Hyehyun Hwang, Chinmoy Sarkar, Boris Piskoun, Naibo Zhang, Apurva Borcar, Courtney L. Robertson, Marta M. Lipinski, Nagendra Yadava, Molly J. Goodfellow, Brian M. Polster

**Affiliations:** 1Department of Anesthesiology and Center for Shock, Trauma and Anesthesiology Research (STAR), University of Maryland School of Medicine, Baltimore, MD 21201, USA; hyehyun.hwang@som.umaryland.edu (H.H.); csarkar@som.umaryland.edu (C.S.); bpiskoun@som.umaryland.edu (B.P.); naibo.zhang@som.umaryland.edu (N.Z.); aborcar@som.umaryland.edu (A.B.); mlipinski@som.umaryland.edu (M.M.L.); nyadava@som.umaryland.edu (N.Y.); mgoodfellow@som.umaryland.edu (M.J.G.); 2Department of Anesthesiology and Critical Care Medicine, Johns Hopkins University, Baltimore, MD 21287, USA; crober48@jhmi.edu

**Keywords:** C1Q, CCI, EFNA, EPHA, mitochondria, neuroinflammation, neuropathology, SHC1, SUZ12, TBI, TRKA

## Abstract

Traumatic brain injury (TBI) leads to persistent pro-inflammatory microglial activation implicated in neurodegeneration. Idebenone, a coenzyme Q10 analogue that interacts with both mitochondria and the tyrosine kinase adaptor SHC1, inhibits aspects of microglial activation in vitro. We used the NanoString Neuropathology Panel to test the hypothesis that idebenone post-treatment mitigates TBI-pathology-associated acute gene expression changes by moderating the pro-inflammatory microglial response to injury. Controlled cortical impact to adult male mice increased the microglial activation signature in the peri-lesional cortex at 24 h post-TBI. Unexpectedly, several microglial signature genes upregulated by TBI were further increased by post-injury idebenone administration. However, idebenone significantly attenuated TBI-mediated perturbations to gene expression associated with behavior, particularly in the gene ontology–biological process (GO:BP) pathways “ephrin receptor signaling” and “dopamine metabolic process”. Gene co-expression analysis correlated levels of microglial complement component 1q (*C1q*) and the neurotrophin receptor gene *Ntrk1* to large (>3-fold) TBI-induced decreases in dopamine receptor genes *Drd1* and *Drd2* that were mitigated by idebenone treatment. Bioinformatics analysis identified SUZ12 as a candidate transcriptional regulator of idebenone-modified gene expression changes. Overall, the results suggest that idebenone may enhance TBI-induced microglial number within the first 24 h of TBI and identify ephrin-A and dopamine signaling as novel idebenone targets.

## 1. Introduction

Microglia, the innate immune cells of the brain, acutely respond to traumatic brain injury (TBI) by proliferating and migrating to the site of damage [1,2]. Once there, the phagocytic cells engulf dying cells and cellular debris that are sources of immune-triggering molecules collectively called damage-associated molecular patterns (DAMPs). This initial microglial response is thought to help restore homeostasis by limiting the magnitude of the broader immune response that includes the infiltration of peripheral macrophages and lymphocytes across a compromised blood–brain barrier [2].

The number of pro-inflammatory microglia peaks at about three days post-TBI in the controlled cortical impact (CCI) mouse model [3] but remains elevated for >8 months post-CCI [4,5]. The extent of this persistent pro-inflammatory activation correlates with the severity of neurological impairments in mice [5,6,7]. This evidence suggests that incomplete resolution of the neuroinflammatory response—and possibly also its initial magnitude—is detrimental in mice. Supporting this interpretation, depleting and repopulating brain microglia at seven days [8] or one month [9] post-TBI by transient inhibition of Colony Stimulating Factor 1 Receptor (CSF1R) survival signaling diminishes the number of pro-inflammatory microglia, reduces neurodegeneration, and improves cognitive outcomes at one to three months post-injury, respectively. These results suggest that improving the resolution of the microglial pro-inflammatory response or tempering its initial magnitude may have therapeutic benefits following TBI.

The synthetic Coenzyme Q10 analogue idebenone suppresses pro-inflammatory changes to microglia in vitro, including attenuation of interleukin-1beta (IL-1β) and nitric oxide release [10,11,12]. The compound also shows anti-inflammatory effects in vivo that are associated with neuroprotection in animal models of Parkinson’s disease, Alzheimer’s disease, and stroke [10,11,12]. Idebenone is clinically safe and is used in many countries to treat the hereditary mitochondrial Complex I disease Leber Hereditary Optic Neuropathy (LHON) [13,14]. Given its established safety in humans and its potential for targeting pro-inflammatory microglia, the primary goal of this study was to provide an initial assessment of whether post-TBI idebenone administration is neuroprotective in an experimental mouse model. The secondary goal was a discovery-based inquiry into the compound’s potential mechanism(s) of action. The NanoString (Seattle, WA, USA) nCounter^®^ Neuropathology Panel was chosen as a readout because it provides rapid, highly quantitative information on gene expression changes that can be used as a proxy for brain injury. We hypothesized that idebenone post-treatment would mitigate TBI-pathology-associated gene expression changes by moderating the inflammatory microglial response to injury.

Consistent with our prediction, idebenone rescued TBI-induced perturbations in the expression of several genes on the Neuropathology Panel, with NanoString gene set analysis identifying “Tissue Integrity”, “Transmitter Synthesis and Storage”, and “Axon and Dendrite Structure” as the most significant scores. However, the data suggest that enhancement rather than attenuation of the acute microglial injury response may contribute to the underlying mechanism. “Ephrin receptor activity”, “neurotrophin receptor binding”, and “dopamine neurotransmitter receptor activity” were the gene ontology molecular function (GO:MF) pathways most significantly affected by idebenone in our acute TBI model, providing new avenues for further investigation.

## 2. Materials and Methods

### 2.1. Controlled Cortical Impact (CCI) Mouse Model and RNA Extraction

This study was approved by the University of Maryland Baltimore Institutional Animal Care and Use Committee (protocol# 0318010). All procedures were consistent with the National Institutes of Health Guide for the Care and Use of Laboratory Animals. Adult male C57BL6/J mice (*n* = 3 per group) were subjected to moderate controlled cortical impact TBI (6 m/s impact velocity, 2 mm deformation depth) or sham control (anesthesia only) as previously described [15]. Idebenone (100 mg/kg) or vehicle (corn oil) was given by intraperitoneal injection at one hour and again at five hours after CCI. Mice were randomly assigned to groups, and the experimenter was blinded to the treatment group. Messenger RNA from the peri-lesion brain cortex or the corresponding brain region in sham controls was extracted using a Qiagen (Germantown, MD, USA) RNA extraction kit at 24 h post-injury for analysis by the NanoString nCounter^®^ Neuropathology Panel. The panel consists of 760 neuropathology-associated genes and 10 housekeeping gene controls.

### 2.2. Gene Set Enrichment Analyses

Gene set enrichment analyses for the sham vs. CCI comparison and the CCI + vehicle vs. CCI + idebenone comparison were carried out on housekeeping-gene-normalized NanoString mRNA data using ROSALIND^®^ (https://rosalind.bio/), with a HyperScale architecture developed by ROSALIND, Inc. (San Diego, CA, USA). One sham + vehicle sample failed nSolver™ quality control (QC) analysis due to high binding density. To enable sham–TBI comparisons, the sham + vehicle and sham + idebenone groups were therefore combined into a single sham control group of *n* = 5 that excluded the QC-failed sample. The following panel-designated housekeeping genes were used for normalization by ROSALIND^®^ based on the geNorm algorithm: *Aars*, *Cnot10*, *Csnk2a2*, *Ccdc127*, *Fam104a*, *Lars*, *Supt7l*, and *Tada2b*. The Benjamini–Hochberg method of estimating false discovery rates (FDR) was used to determine adjusted *p*-values (*p*-adj) for pathway analysis using a false discovery rate of 0.05. A *p*-adj of <0.05 was considered significant for gene set enrichment analysis. NanoString cell type profiling was carried out using ROSALIND^®^, which quantifies a given cell population using marker genes that are typically stably expressed and cell-type-specific. The algorithm includes a dynamic selection that identifies the subset of signature marker genes that are robustly expressed. For the discovery-based component of the study, differential gene expressions for the sham vs. CCI and the CCI + vehicle vs. CCI + idebenone comparisons were determined by ROSALIND^®^ using a *p*-value of 0.05 or 0.1, as indicated. To test for significant changes in individual gene transcripts across groups, one-way ANOVA followed by Tukey’s post hoc analysis were conducted using the GraphPad Prism 10.2.2 software (Boston, MA, USA). *p*-values < 0.05 were considered significant. *Z*-scores were calculated from the ROSALIND^®^-normalized data as previously described [16], and heatmaps were generated using GraphPad Prism. Functional enrichment analysis for the GO and Kyoto Encyclopedia of Genes and Genomes (KEGG) pathways was performed using g:Profiler (version e111_eg58_p18_f463989d) with the g:SCS multiple testing correction method, applying a significance threshold of 0.05 [17]. All normalized gene expression data are available in this article and Appendix A, and the raw data files will be deposited into Gene Expression Omnibus (GEO).

### 2.3. Development of the Microglia Gene Signature

To evaluate the effect of idebenone on the microglial TBI response, we developed a microglia gene signature within the Neuropathology gene expression Panel based on analysis of a published dataset (GEO; GSE160651). Genes showing a reduction in expression of ≥70% in the cortical tissue of mice pre-treated with the microglia-depleting drug PLX5622 relative to a vehicle for both the sham and 24 h TBI groups were included in the microglia gene signature set.

## 3. Results

### 3.1. Idebenone Reverses TBI-Induced Gene Expression Changes Related to Tissue Integrity, Transmitter Synthesis and Storage, and Axon and Dendrite Structure Without Broadly Inhibiting the TBI Neuropathology Gene Signature or Cytokine Response

Either idebenone or a vehicle control were administered at one and five hours after moderate TBI induced by controlled cortical impact. Peri-contusional brain cortical tissue was harvested from the TBI or sham control mice, and the NanoString Neuropathology Panel was then used to evaluate whether idebenone modified pathology-associated gene expression changes. Over 100 genes were differentially expressed by ≥1.5-fold following TBI relative to the sham, consistent with the ability of the panel to detect brain injury via gene signatures. The top 15 upregulated and downregulated genes following TBI compared to the sham are shown in Figure 1a. These genes all showed a >6-fold increase or a >2-fold decrease in the TBI cohort, respectively. NanoString gene set analysis identified “Cytokines” as the most significantly perturbed gene signature, followed by “Neuronal Cytoskeleton”, “Activated Microglia”, “Angiogenesis”, “Disease Association”, “Matrix Remodeling”, “Oxidative Stress”, “Apoptosis”, “Growth Factor Signaling”, and “Tissue Integrity”. A list of significantly altered genes belonging to each NanoString signature is given in Appendix A. Normalized mRNA counts from the entire dataset are given in Appendix A.

Idebenone did not broadly inhibit the acute (24 h) TBI neuropathology gene signature (Figure 1a), nor did it suppress the cytokine response (Figure 1b). However, select gene alterations resulting from TBI were significantly affected by idebenone treatment (*p* < 0.05), including several of the genes most susceptible to TBI-induced downregulation (Figure 1a and Table 1). In total, 43 genes were differentially expressed at *p* < 0.05 between the TBI + vehicle and the TBI + idebenone group (Table 1). The significance for *Npas4* (not listed) was driven by an outlier (Appendix A), and this gene was excluded from the gene ontology enrichment analysis.

Most gene expression differences in the TBI + idebenone group represented reversals of TBI-induced gene expression changes (Figure 2a). “Tissue Integrity”, “Transmitter Synthesis and Storage”, and “Axon and Dendrite Structure” were the highest-scoring results of the NanoString gene set analysis, suggesting that idebenone may reduce the loss of select cell types or synapses following TBI. The top 10 NanoString gene set significance scores are shown in Table 2, along with the genes belonging to each category.

Relaxing the significance criteria to *p* < 0.1 identified an additional five genes altered by >1.5-fold for the TBI + vehicle vs. TBI + idebenone comparison (Table 3). Idebenone also reversed TBI-induced changes in these genes (Figure 2a). Functionally, these genes all belonged to one or more of the top-scoring signatures identified by the NanoString gene set analysis (Table 2), suggesting that they may be legitimate targets of idebenone treatment. The list of all genes altered at a significance level between *p* > 0.05 and *p* < 0.1, irrespective of fold change, is provided in Appendix A.

### 3.2. Idebenone Enhances Expression of Microglia Signature Genes Following TBI

We predicted that idebenone would decrease the inflammatory microglial response to injury. Using the NanoString Cell Type Profiler algorithm, we found that TBI significantly increased microglial and endothelial marker genes. There was not a clear effect of idebenone on the TBI microglial response (or on endothelial cells) based on these markers. However, CCI compromises the blood–brain barrier, leading to the brain entry of peripheral immune cells that appear to be the main initial drivers of the pro-inflammatory immune response [7]. Infiltrating macrophages express some of the same genes as microglia, potentially obscuring the microglial gene signature following TBI. To more rigorously employ gene expression changes to evaluate whether idebenone alters the TBI-induced microglial response, we sought to identify which genes on the Neuropathology Panel were selectively expressed by microglia both under normal conditions (sham injury control group) and following TBI.

Microglia survival in vivo depends on constitutive signaling through colony-stimulating factor receptor 1 (CSFR1) [19]. Accordingly, one-week administration of CSFR1 antagonist depletes the brain of >95% of microglia in mice [19,20]. Previously, the Neuropathology Panel was used to investigate cortical gene expression at 24 h after mouse TBI, both with and without prior microglia depletion by the CSFR1 antagonist PLX5622 [20]. We reanalyzed this dataset (GEO; GSE160651) to define a microglial gene signature comprised of the subset of Neuropathology Panel genes showing ≥70% depletion following PLX5622 treatment, both with and without prior TBI (Figure 3a). Most genes that displayed predominantly microglial expression in uninjured animals were also strongly depleted by PLX5622 following TBI. However, *Stabilin-1* (*Stab1*) did not meet the signature set criteria (only ~60% PLX5622-induced depletion following TBI) despite still showing microglia-biased expression.

In our CCI model, we found that, as expected, the expression of several activation-associated microglial genes significantly increased following TBI, whereas the expression of the homeostatic microglial genes *P2ry12* and *Tmem119* decreased (*p* < 0.0001) or trended down (*p* = 0.065), respectively (Figure 3b). Contrary to our hypothesis, several microglial signature genes that were upregulated by TBI, e.g., *Ccr5*, *C1qb*, and *Fcrls*, were further increased by the idebenone post-TBI treatment (*p* < 0.05), whereas no microglial signature genes were decreased by idebenone (Figure 3b). Idebenone similarly increased expression of the microglia-biased gene *Stab1* following TBI (Table 1).

The literature indicates that a disease-associated microglial subtype expressing TPSO and NOX2, encoded by *Tpso* and *Cybb*, respectively, is pathogenic following TBI [21,22]. Although idebenone treatment led to an overall increase in the microglial signature following TBI (suggesting an increased microglial number), neither *Tpso* nor *Cybb* mRNA were increased by idebenone (Figure 4). In addition, idebenone did not alter the 24 h post-TBI expression level of genes encoding NLRP3 inflammasome components NLRP3 or caspase-1. These results suggest that idebenone promotes healthy rather than neurotoxic microglial accumulation.

### 3.3. “Ephrin Receptor Signaling” and “Dopamine Metabolic Process” Are Top Gene Ontology Pathways Affected by Idebenone Treatment

To further explore the effects of idebenone treatment on TBI, we performed additional gene set enrichment analysis using g:Profiler [17]. This analysis identified “ephrin receptor signaling pathway” and “dopamine metabolic process” as the most significant specific gene ontology biological processes (GO:BP) affected by idebenone treatment (Figure 2b,c), along with the broad “cell-cell signaling” and “behavior” GO:BP pathways. “Ephrin receptor activity”, “signaling receptor binding”, “neurotrophin receptor binding”, and “dopamine neurotransmitter receptor activity” were the most significant gene ontology molecular function (GO:MF) pathways (Figure 2d). The “cAMP signaling pathway”, which plays a central role in neurotransmitter-dependent G-protein-coupled signal transduction, was the most significantly enriched KEGG pathway (Figure 2e). Consistent with potential effects of idebenone on cell–cell signaling and neurotransmission, gene ontology cellular component (GO:CC) analysis identified the somatodendritic compartment (GO:0044297; p-adj < 7.738 × 10^−11^) and cell body (GO:0044297; p-adj < 3.282 × 10^−11^) as the cellular compartments highly affected by post-TBI idebenone treatment.

### 3.4. Idebenone Mitigates Widespread TBI-Induced Perturbations to Ephrin-A Signaling Genes

Because gene set enrichment analysis identified ephrin and dopamine signaling as idebenone targets following TBI, we evaluated changes to these pathways in more detail. Ephrin receptors are tyrosine kinase receptors that can interact with multiple membrane-bound ephrin-A ligands but with differential binding affinities. Expression levels of ephrin-A ligand and receptor genes were broadly dysregulated in peri-lesional brain cortical tissue at 24 h post-injury (Figure 5a). TBI upregulated the ephrin-A1 ligand (EFNA1)-encoding gene *Efna1* (Figure 5a,b) and multiple ephrin-A receptor-encoding genes (*Epha2*, *Epha5*, and *Epha6*, Figure 5a,c). Conversely, *Efna5*, which encodes ephrin-A5 ligand (EFNA5, Figure 5a,b), and *Epha4*, which encodes the EPHA4 receptor with a higher affinity for EFNA5 than EFNA1 (Figure 5a,c), were both downregulated following TBI. Idebenone mitigated TBI-induced gene expression changes to both ephrin-A ligand-encoding (*Efna1* and *Efna5*) and receptor-encoding (*Epha5* and *Epha6*) genes while also downregulating *Epha3* and *Epha7* in the context of TBI (Figure 5a–c).

### 3.5. Idebenone’s Ability to Rescue the Gene Expression Levels of Select GO:BP Dopamine Metabolic Process Genes Following TBI Reveals an Idebenone Partial Responder

Dopamine receptor D2 (*Drd2*) exhibited the greatest fold-change for the TBI + idebenone to TBI + vehicle comparison (Table 1). However, greater variability in *Drd2* expression was seen compared to the expression of ephrin-A signaling genes, with idebenone failing to protect against the TBI-induced *Drd2* reduction in one individual (Figure 6a). To facilitate comparison of gene expression patterns in the TBI groups, we transformed the data from the six CCI-injured animals into a *z*-score, which compares each individual value to the mean value across the injured group, irrespective of vehicle or idebenone treatment. Indeed, although *Efna5* expression, like *Drd2* expression, was decreased by TBI and rescued by idebenone post-treatment (Figure 6b), the expression level of *Efna5* (Figure 6b, note open circles) and other ephrin-A pathway genes (Figure 6c) did not correlate with *Drd2* expression level in brain-injured animals. However, upon examination of genes in the GO:BP dopamine metabolic pathway, we found that dopamine receptor D1 (*Drd1*) and phosphodiesterase 1b (*Pde1b*) showed a very similar pattern of gene expression variability to *Drd2* in the TBI groups (Figure 6a, note open circles). Linear regression analysis confirmed that the *Drd1* and *Pde1b* expression patterns highly correlated with that of *Drd2* (Figure 6d, R^2^ = 0.98 and R^2^ = 0.96, respectively, both *p* < 0.001), with idebenone failing to rescue TBI-induced *Drd1* and *Pde1b* mRNA decreases in the same animal.

Interestingly, this variability in idebenone response parallels the clinical situation; only ~50% of LHON mitochondrial disease patients show improvement upon idebenone treatment [23,24]. Idebenone’s behavior as a pro-drug, requiring reduction to idebenol to act as an antioxidant and mitochondrial electron donor [13,25,26], may contribute to its variable efficacy in vivo. This possibility is supported by recent evidence that inactivating single-nucleotide polymorphisms in the gene that encodes the idebenone-reducing enzyme NAD(P)H quinone dehydrogenase 1 (NQO1) are associated with idebenone non-responders [24]. In our study, we found that although the individual mouse showing failed *Drd2* rescue by idebenone had the lowest level of *Nqo1* mRNA (Figure 6b), the *Nqo1* level did not significantly correlate with the *Drd2* level in the TBI + idebenone group (Figure 6c, R^2^ = 0.2452, *p* = 0.67). This finding suggests that other, or additional, factors explain the variable *Drd2* expression.

The dopamine metabolic pathway genes are a subset of the broader GO:BP “behavior” pathway (Figure 2c). We evaluated gene expression variability within this larger pathway, seeking to determine whether the low-*Drd2* individual could be classified as an idebenone partial/non-responder, and if so, whether the variability in response could be exploited to gain mechanistic insight into neurotransmitter-/behavior-related processes through gene co-expression analysis. We were indeed able to identify additional genes in the “behavior” pathway, e.g., *Adora2a* and *Adcy5*, that showed a very similar pattern of changes to *Drd2* (Figure 6a,d, R^2^ = 0.99 and R^2^ = 0.95, respectively, both *p* < 0.001). However, we found that not all dopamine-related or neurotransmitter-related genes exhibited the same pattern of alterations. For example, dopamine receptor D4 (*Drd4*) was upregulated by TBI but was unchanged by idebenone treatment, whereas the gene that encodes the neuromodulator Neuropeptide Y (*Npy*), was downregulated in the idebenone post-TBI treatment group but was not significantly decreased by TBI alone (Figure 6b). Furthermore, the expression of both these genes exhibited a very different pattern of variability across individuals than *Drd2* (Figure 6b compared to Figure 6a, note open circles). Catechol-O-methyltransferase (*Comt*), which encodes an enzyme responsible for dopamine, epinephrine, and norepinephrine degradation, was upregulated by TBI but showed a further increase with post-injury idebenone treatment (Figure 6b), much like the pattern observed for microglial signature genes (Figure 3b). Notably, although the *Comt* mRNA level for the TBI + idebenone group represented only a 1.16-fold increase compared to TBI + vehicle, the difference was still highly significant (*p* < 0.001) owing to the low variability across individual mice. These data indicate that all mice receiving idebenone showed some response to the drug, i.e., it is unlikely that the *Drd2* gene expression outlier was due to failed idebenone administration. Linear regression analysis confirmed that there was no significant correlation between *Drd4*, *Npy*, or *Comt* expression and *Drd2* expression for mice that received TBI (Figure 6e). These results, when considered alongside the many significant idebenone-dependent gene expression differences seen across the ephrin-A pathway (Figure 5), indicate that the *Drd2* outlier individual can be considered a partial responder to a compound with potentially multiple mechanisms of action.

### 3.6. Complement Component 1q (C1q) and Homeostatic Microglial Gene Expression Levels Correlate with Expression of a GO:BP “Behavior” Gene Subgroup Following TBI

Functional neuroprotection is the desired goal of idebenone treatment. Therefore, we used gene co-expression analysis to see whether we could identify additional signaling processes or cell-specific markers that correlate with the potentially protective gene expression changes to the GO:BP “behavior” pathway. First, as idebenone appeared to enhance the early microglial response to TBI, we used Pearson’s correlation analysis to ask whether there was a relationship between idebenone’s effect on microglial signature genes and its effect on the “behavior” pathway. We found a strong correlation between complement component 1q (*C1q*) genes and the subset of *Drd2*-correlated genes within the “behavior” pathway (Figure 7a). Genes considered markers of homeostatic microglia (e.g., *P2ry12* and *Tmem119*) but not microglial genes typically associated with activated/disease-associated states (e.g., *Ncf1*) also correlated with the *Drd2*-related “behavior” gene subgroup (Figure 7a). In contrast, the microglial signature gene expression pattern did not correlate as strongly with that of ephrin-A signaling genes (Figure 7b). However, homeostatic microglial genes did show a modest inverse correlation with *Epha2*, while several activated/disease-associated microglial genes trended toward a positive correlation with *Efna5* and a negative correlation with *Efna1* (most showing *p*-values between 0.05 and 0.1). These results are consistent with the possibility that homeostatic microglia and C1Q signaling play a role in idebenone’s effects on the *Drd2*-related “behavior” gene subgroup, whereas idebenone’s enhancement of the acute microglial TBI response and effects on ephrin-A signaling gene expression may be mediated by separate mechanisms.

### 3.7. Idebenone Alters Expression of Many Genes Encoding SHC1-Interacting Tyrosine Kinase Receptors

Idebenone was reported to bind Shc adaptor protein 1 (SHC1) at concentrations below those necessary for stimulating mitochondrial electron transport [27]. SHC1 binding may underlie idebenone’s modulation of ephrin-A signaling gene expression, as SHC1 is a protein that acts as a scaffold to couple tyrosine kinase activation—including that mediated by ephrin-A receptors [28,29]—to downstream signaling. Given this possibility, we hypothesized that idebenone broadly affects receptor tyrosine kinase gene expression by altering tyrosine kinase signaling through SHC1 interaction. The NanoString Neuropathology Panel includes 16 members of the GO:MF pathway ”transmembrane receptor protein tyrosine kinase activity” (GO:0004714), and of those, the majority are known to interact with SHC1. Consistent with our hypothesis, we identified receptor tyrosine kinase proteins outside the ephrin-A family that were significantly altered by TBI and/or idebenone treatment (Figure 8a). A TBI-induced decrease in *Ntrk1*, which encodes the pro-survival nerve growth factor receptor tropomyosin receptor kinase (TRKA), was attenuated by post-TBI idebenone administration (Figure 8a,b). Additionally, insulin-like growth factor 1 receptor (*Igf1r*), known to promote pro-survival signaling, showed increased expression when idebenone was given post-TBI but was not elevated by TBI in the absence of idebenone treatment (Figure 8a,b). Both TRKA [30,31] and IGF1R [32] are known to interact with SHC1. Three non-receptor tyrosine kinase genes on the panel, *Abl*, *Lrrk2*, and *Src*, encode enzymes also reported to interact with SHC1 [33,34,35,36,37]. Of these, two appeared to respond to idebenone treatment. Idebenone mitigated a TBI-induced decrease in *Lrrk2* (*p* < 0.05), similarly to its rescue of *Ntrk*, while *Src* expression was higher in the TBI + idebenone group (*p* < 0.1), but like *Igf1r*, it was not increased by TBI in the untreated animals (Figure 8c). Together with the results of Figure 5 showing idebenone’s effects on the ephrin-A signaling pathway, these results are consistent with the possibility that idebenone engages SHC1 in vivo to preserve or promote tyrosine kinase gene expression following TBI.

### 3.8. Expression of TRKA-Encoding Ntrk1, but Not That of Idebenone-Altered Growth Factor Genes, Correlates with Drd2-Related “Behavior” Gene Subgroup Changes Following TBI

Next, given that the pattern of ephrin-A pathway gene expression following TBI failed to correlate with that of the *Drd2*-correlated “behavior” gene subgroup, we considered the possibility that a SHC1-binding tyrosine kinase outside the immediate ephrin-A signaling pathway may be mechanistically related to the “behavior” pathway gene expression changes and idebenone action. To begin to evaluate this possibility, we performed gene co-expression analysis using Pearson’s correlation of the *Drd2*-correlated “behavior” subgroup and the other tyrosine kinase genes on the panel. Among the tyrosine kinase genes, only the expression levels of the TRKA-encoding gene *Ntrk1* and the homeostatic microglial gene *Csf1r* significantly correlated with the expression of both *Drd1* and *Drd2* (Figure 9a, *p* < 0.05). *Igf1r* expression correlated with *Drd1* and *Pde1b* expression while showing a trend (*p* = 0.06) toward *Drd2* expression correlation, whereas *Lrrk2* correlated with *Pde1b* expression while showing a trend toward *Drd1* and *Drd2* correlation (*p* = 0.06 and *p* = 0.08, respectively). *Ntrk1* belongs to the GO:MF “neurotrophin receptor binding pathway”, which was altered by idebenone treatment. Growth factor receptor ligands neurotrophin 3 (*Ntf3*), hepatocyte growth factor (*Hgf*), and fibroblast growth factor 14 (*Fgf14*) were all identified as idebenone TBI targets (Table 1). This prompted us to investigate whether other neurotrophin signaling molecules or growth factors are correlated with the pattern of “behavior” gene expression changes following TBI. However, upon co-expression analysis, we found that *Ntrk1* was unique among the Neuropathology Panel growth factor/neurotrophin signaling genes in showing a correlation with the *Drd2*-related “behavior” gene subgroup (Figure 9b). These results suggest that none of the growth factor receptor ligands altered by post-TBI idebenone treatment are mechanistically linked to the large changes in dopamine receptor gene expression that occur after TBI.

### 3.9. Bioinformatics Analysis Identifies Additional Co-Regulated Genes and the Transcriptional Regulator SUZ12 as Candidate Mediators of Idebenone-Modified TBI-Induced Gene Expression Changes

Finally, we used the most altered gene in the TBI + idebenone cohort, *Drd2*, to perform a correlation analysis across the entire panel, seeking to identify additional signaling genes or processes that correlated with the “behavior” gene subgroup showing TBI-impaired expression rescued by idebenone treatment. Pearson’s correlation analysis identified several genes encoding cytokines, receptors, or signaling molecules that showed expression levels inversely correlating with *Drd2* expression. These included *Cxcr4* (*r* = −0.8635, *p* < 0.05), *Grm2* (*r* = −0.8391, *p* < 0.05), *Cxcl12* (*r* = −0.8359, *p* < 0.05), *Tenm2* (*r* = −0.7869, *p* = 0.06), and *Mmp9* (*r* = −0.7583, *p* = 0.08), which are shown in Figure 10a. As these genes demonstrated higher expression in the brains of animals with reduced levels of *Drd2*-related “behavior” subgroup mRNAs, they can be considered candidate mediators of neurobehavioral impairment. Meanwhile, we found several genes related to acetylcholine or γ-aminobutyric acid (GABA) signaling (Figure 10b) and a group of myelin-related oligodendrocyte-expressed genes (Figure 10c) that showed a TBI expression pattern linearly correlating with *Drd2* (all *p* < 0.05). Because the NanoString panel measures bulk mRNA expression changes rather than quantifying mRNA changes in single cells, one interpretation of the results depicted in Figure 10b,c is that the TBI-induced gene expression changes represent a loss of specific cell types, e.g., interneurons or oligodendrocytes, that were rescued by idebenone in the full, but not partial, drug responders. An alternative (but not mutually exclusive) possibility is that TBI and idebenone reciprocally modified cortical gene transcription in the CCI animal model, contributing to or causing many of the observed gene expression alterations.

To provide insight into the likelihood of this second possibility, we conducted further enrichment analysis of the gene set altered by idebenone in the context of TBI (see Figure 2a) using the “ENCODE and ChEA_Consensus TFs from ChIP-X” library in Enrichr [38,39,40]. This approach identified polycomb repressive complex 2 subunit SUZ12 as a candidate regulator of the expression of idebenone-changed genes (*p*-adj < 0.00001, Figure 10d). Including the co-expressed genes described in Figure 10a–c when conducting the Enrichr gene set enrichment analysis yielded similar results, with SUZ12 far outstripping other candidates (*p*-adj < 1 × 10^−9^ vs. *p*-adj = 0.012 for the next closest candidate, ESR1, Appendix A). *Cxcl12*, *Cxcr4*, *Gabra4*, *Gad2*, *Mal*, *Mmp9*, and *Slc32A1* were identified as *Drd2*-correlated genes that may also be regulated by SUZ12. The next three candidate transcriptional regulators showed marginal significance (Figure 10d, *p* < 0.05, *p*-adj < 0.1) and are predicted to target fewer of the idebenone-altered gene set. However, it is notable that among the eight genes that may be regulated by REST were *Drd2* and five genes showing *Drd2*-correlated gene expression following TBI (the positively correlated *Adcy5* and *Drd1* and the inversely correlated *Grm2*, *Htr1a*, and *Tenm2*). The other top candidates for mediating idebenone-dependent gene expression changes following TBI—estrogen receptor 1 (ESR1) and androgen receptor (AR)—were sex-hormone-responsive transcriptional regulators. Genes that may be AR-regulated following TBI ± idebenone treatment included *Drd2* and several ephrin-A signaling genes of the top-scoring GO pathways, as well as the idebenone-rescued *Dgkb* (Figure 10e). The anti-apoptotic gene *Bcl-2* stood out among the genes that may be ESR1-regulated, given that “tissue integrity” was the most significant score of the NanoString gene set analysis. TBI triggered downregulation of *Bcl-2* expression that was mitigated by idebenone post-treatment (Figure 10f).

## 4. Discussion

Idebenone post-treatment significantly ameliorated many gene expression changes in the peri-lesional cortex caused by TBI at an acute, 24 h post-injury timepoint. The similarity of the TBI cytokine response and the top differentially expressed genes between the vehicle- and idebenone-treated groups indicates that the gene expression differences in the idebenone-treated cohort cannot be explained by an inconsistent level of CCI injury across groups. The expression of several microglial signature genes significantly elevated by TBI was increased further by idebenone post-injury administration (*p* < 0.05), which likely simply reflects an increased number of phagocytic, debris-clearing microglia in the drug-treated animals. However, because the other genes identified by differential gene expression/co-regulation analysis are typically expressed by distinct cell types, e.g., interneurons, oligodendrocytes, and endothelial cells, it is unlikely that loss of a single cell type explains the gene expression deficits that are rescued by idebenone treatment. Changes to genes in the ephrin-A signaling pathway, as well as to those encoding dopamine receptors (e.g., *Drd1* and *Drd2*), and their downstream signaling components (e.g., *Pde1* and *Adcy5*) may instead occur through specific mechanisms. A bioinformatics search for transcriptional regulators reinforces this possibility, and gene co-regulation analysis suggests that idebenone affects gene expression through at least two different mechanisms.

Our differential gene expression findings, combined with the prior identification of idebenone as a SHC1 inhibitor [27], led us to hypothesize that as one of the mechanisms, the compound modifies in vivo receptor tyrosine kinase gene expression by altering SHC1-dependent tyrosine kinase signaling. In addition to effecting widespread changes in ephrin-A signaling gene expression, idebenone significantly altered the expression of additional tyrosine kinase receptor genes and genes encoding their ligands. As idebenone is predicted to engage SHC1-containing tyrosine kinase complexes at the protein level, the mechanisms by which idebenone modulates the mRNA transcripts encoding the affected kinases are not clear but may include activity-responsive transcription factors. Bioinformatics analysis identified the epigenetics modulator SUZ12 as a top transcriptional regulator candidate, followed by ESR1, REST, and AR. The top candidate, SUZ12, is essential for histone 3 lysine 27 methylation by the Polcomb repressive complex 2/3 [41,42,43]; however, its roles in brain injury and neuroprotection have yet to be explored.

Idebenone’s mitochondrial effects are a second potential mechanism of action lying upstream of its gene expression effects. In cell types that can enzymatically convert idebenone to its reduced form, idebenol, the drug can bypass mitochondrial Complex I deficits in vitro, stimulating respiration by transferring electrons to Complex III [25,26,44]. The idebenol form also acts as an antioxidant to inhibit lipid peroxidation, including lipids of both the mitochondrial and plasma membranes [13,45,46,47,48]. Whether the mitochondrial or antioxidant actions of idebenone are required for its ability to affect post-TBI gene expression changes and, possibly, microglial number is not clear. However, one mouse showed an incomplete response to idebenone in the expression of several neurotransmitter-related genes (e.g., *Drd1* and *Drd2*), just as idebenone non-responders are found among LHON patients. Evidence that single-nucleotide variants encoding the idebenone-reducing enzyme *NQO1* correlate with idebenone responsiveness in LHON [24] supports the hypothesis that idebenone’s mitochondrial and antioxidant mechanisms of action in vivo require bioactivation. Failed idebenone bioactivation may also explain the incomplete gene expression response in our study, linking those genes to idebenone’s second mechanism of action.

Our study identified ephrin-A signaling and dopamine signaling as the top molecular pathways affected by the idebenone intervention after TBI. There are only a few published reports on ephrin-A pathways in the context of TBI. An *Epha6* mouse knockout study suggests that EPHA6 promotes cell death and astrogliosis following TBI [49]. We found significantly increased *Epha6* mRNA in peri-contusional brain cortical tissue following TBI that was completely prevented by idebenone post-treatment, consistent with a protective effect. EPHA4 is increased at the protein level in reactive astrocytes following human TBI [50], and the rodent literature suggests that EPHA4, like EPHA6, contributes to brain injury [51,52,53]. In our study, *Epha4* mRNA decreased following TBI, and this decrease was not modified by idebenone treatment. However, more interesting results were observed related to EPHA2 signaling. TBI is a risk factor for post-traumatic stress disorder (PTSD), and multiple single-nucleotide variants of *EPHA2* are linked to PTSD [54]. *EPHA2* expression is increased in PTSD brains [55], and we found that *Epha2* mRNA was significantly increased 24 h after TBI. TBI also increased *Efna1*, which encodes a preferred ligand for EPHA2 compared to *Efna5*, which was decreased after TBI. Although idebenone treatment did not prevent the increase in *Epha2* mRNA, idebenone significantly reversed the TBI-induced perturbations to *Efna1* and *Efna5*, which may relate to PTSD risk.

In contrast to ephrin-A signaling, dysregulation of dopaminergic signaling after TBI is well-documented [56]. However, most studies have focused on the striatum rather than the cortex. Nevertheless, a single-cell RNA sequencing study found that in cortical neurons, suppression of dopamine signaling was one of the most significantly altered pathways following TBI at seven days post-injury [20]. The same investigators detected a decrease in both *Drd1* and *Drd2* by the NanoString Neuropathology Panel at 30 days post-TBI [20]. We found large decreases in both *Drd1* and *Drd2* gene expression at 24 h post-injury, indicating that the disruption to dopamine receptor gene expression occurs acutely and, based on the literature, appears to be sustained. *Drd1* and *Drd2* were among the four genes exhibiting the greatest TBI-induced reduction in expression that were rescued by idebenone administration (Table 1).

In addition to providing insight into idebenone’s in vivo mechanisms of action, our unique dataset offers insight into the early pathogenesis of TBI. The magnitude of the TBI-induced decline in *Drd1* and *Drd2* expression, combined with the critical signaling roles of the co-regulated genes, suggests that the TBI-triggered decrease in the “behavior” gene subgroup is deleterious. We wondered whether specific cytokines or receptors might drive the TBI-induced downregulation to the *Drd2*-related “behavior” gene subgroup and, therefore, show TBI-induced changes in gene expression that inversely correlate with that of *Drd2*. From our gene co-regulation analysis, *Cxcl12*, *CxCr4, Mmp9*, *Tenm2*, and *Grm2* emerged as the top candidates, and most of these have already been linked to TBI.

Among these, *Cxcl12* and *CxCr4* stand out, as dysregulation of the CXCL12 ligand/CXCR4 receptor axis has been implicated in the cognitive deficits associated with several neurodegenerative diseases [57]. Among their activities, CXCL12/CXCR4 mediate inflammation through monocyte recruitment [58]; CXCR4 antagonism attenuates inflammatory cytokine expression, which may diminish the influence of neuroimmune signaling on synaptic plasticity and other processes that underlie cognition and emotion [59,60,61]. Elevated serum CXCL12 concentration in TBI victims is highly associated with trauma severity and mortality [62]. However, increased CXCR4 and CXCL12 mRNAs in peri-contusional brain tissue correlate with a trend toward more favorable TBI clinical outcomes [63], indicating a need for further research.

*Mmp9* is also an intriguing result of our analysis. As an enzyme that can degrade the blood–brain barrier, matrix metalloproteinase-9 (MMP9) upregulation can increase edema, decrease cerebral perfusion, and allow for greater peripheral immune cell infiltration, thereby promoting inflammation [64]. These factors may contribute to greater secondary brain injury. MMP9 inhibition is already being investigated as a therapeutic mechanism for TBI [64].

Little is known about the role of teneurin-2 (*Tenm2*) in TBI. It was identified as a transmembrane protein found in cortical pyramidal neurons in the healthy brain [65], putatively as a presynaptic cell adhesion molecule [66], which may make it important for plasticity. Brain injury acutely increases teneurin-2 expression in reactive astrocytes, but its role in injury processes is unknown [65].

Finally, metabotropic glutamate receptor 2 (mGluR2), encoded by *Grm2*, is found on neurons, microglia, oligodendrocytes, and astrocytes. Changes in mGluR2 expression are associated with neurodegenerative diseases, as well as schizophrenia, drug addiction, and depression [67]. In TBI, mGluR2 is linked to upregulation of gap junctions, which may contribute to cell death [68]; antagonism may limit secondary brain injury and preserve function [69].

We also used gene co-expression analysis to ask whether any signaling genes, such as those encoding neurotrophin receptors, positively correlate with the *Drd2*-related “behavior” gene subgroup, potentially participating in idebenone’s mechanism of action. We discovered that expression of complement component 1q (*C1q*) and the neurotrophin receptor gene *Ntrk1* both correlated with TBI-induced decreases in dopamine receptor genes *Drd1* and *Drd2* that were mitigated by idebenone treatment. Further experiments are necessary to determine whether either C1Q or TRKA signaling contribute to idebenone’s ability to modify cortical gene expression in vivo.

Our study has several important limitations. One is the small sample size. This study was undertaken as a first-pass investigation to determine whether more extensive evaluation of idebenone as a neuroprotective agent in TBI is warranted. Therefore, while idebenone’s effects on neuroinflammation and TBI recovery cannot be evaluated for durability or functional significance because this study is restricted to a single acute time point, this study was intentionally of a limited scope. While supporting further investigation, the results suggest that longitudinally investigating idebenone’s effects at later time points, e.g., those showing peak or persistent microglial activation and/or long-term behavioral deficits, would be more informative from a cost/benefit standpoint than extending study of the acute time point. A second limitation is that only male mice were studied. Sex differences following TBI are reported, and both sexes will be included in follow-up investigations. Given that two of the four candidate idebenone-responsive transcriptional regulators, ESR1 and AR, are sex-hormone-responsive transcription factors, it is possible that sex will affect not only TBI outcome but also the efficacy of idebenone treatment. A third limitation is that bulk gene expression analysis was carried out using tissue samples containing multiple cell types. This may overestimate, underestimate, or obscure changes occurring in individual cell types. For this reason, it is important to note that even relatively minor fold-changes in gene expression seen with idebenone treatment may be biologically significant if they reflect larger changes occurring in a single cell type. Future studies could include investigation of differential gene expression by single-cell RNA sequencing or spatial techniques.

## 5. Conclusions

Overall, the results from this pilot transcriptomics study support the possibility that idebenone may be functionally neuroprotective and motivate a more extensive evaluation of idebenone for TBI treatment. It is important to note that our results provide only circumstantial evidence that idebenone can engage and signal either through SHC1- or mitochondria-dependent mechanisms in vivo; additional experiments are needed to test these hypothesized mechanisms of action. Similarly, while an increase in microglial number is the most likely explanation for the increased expression of microglial signature genes, complementary validation, e.g., by immunophenotyping or cell-type-specific labeling, should be carried out to support this conclusion. The persistent activation of microglia is hypothesized to be a factor in the progressive clinical symptoms and pathology displayed in chronic TBI. From a therapeutic development perspective, it will be important to determine the long-term effects of idebenone treatment on microglial reactivity to discern if the compound has potential for TBI treatment.

## Figures and Tables

**Figure 1 cells-14-00824-f001:**
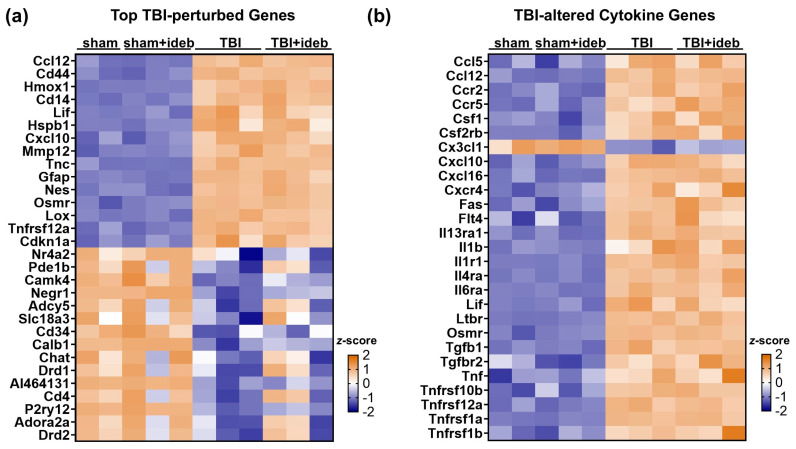
Heatmaps showing the expression levels across all samples of (**a**) the most altered genes following traumatic brain injury (TBI) and (**b**) the significantly altered cytokine-encoding genes following TBI, with relative transcript levels represented by *z*-score. Relative to the mean value across groups for a given gene, a more orange color indicates a relative increase in gene expression, whereas a bluer color indicates a relative decrease. Sham samples containing either idebenone or vehicle are shown separately to allow for comparison, although they were combined into a single sham group for differential gene expression analysis.

**Figure 2 cells-14-00824-f002:**
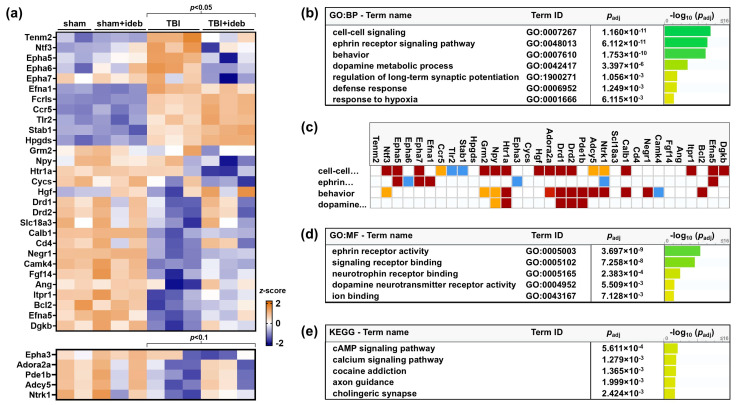
“Ephrin receptor signaling” and “dopamine metabolic process” are the top TBI-perturbed gene ontology (GO) pathways altered by post-injury idebenone treatment. (**a**) Heatmap showing relative expression of the top idebenone-altered genes across all samples. Relative to the mean value across groups for a given gene, a more orange color indicates a relative increase in gene expression, whereas a bluer color indicates a relative decrease. (**b**) Top GO–biological process (GO:BP) pathways affected by idebenone treatment. (**c**) Map of idebenone-altered genes belonging to the top four GO:BP pathways shown in b. The g:Profiler color codes reflect the strength of evidence for the indicated pathway, from strong (dark red = direct assay; medium red = genetic interaction or physical interaction) to modest (orange = sequence or structural similarity or genomic context) to weak (blue = reviewed computational analysis or electronic annotation). (**d**) Top GO–molecular function (GO:MF) pathways affected by idebenone treatment. (**e**) Top Kyoto Encyclopedia of Genes and Genomes (KEGG) pathways affected by idebenone treatment.

**Figure 3 cells-14-00824-f003:**
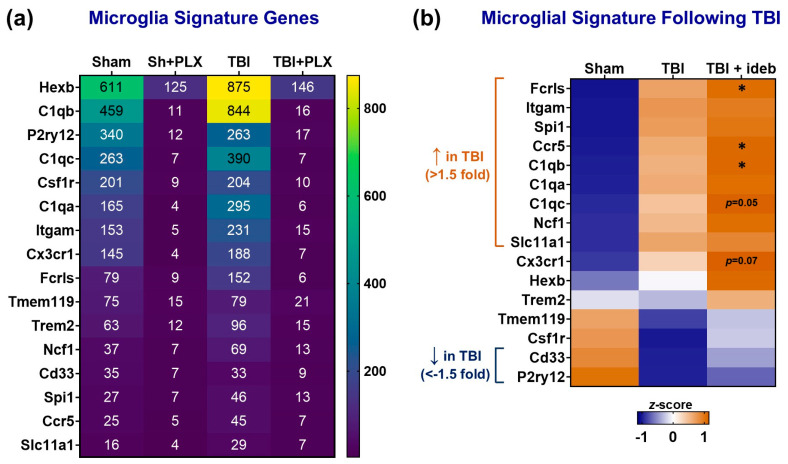
Idebenone increases microglial signature gene expression following TBI. (**a**) Heat map defining the microglial gene signature set based on mean normalized mRNA counts in brain cortical tissue obtained from sham or TBI mice receiving vehicle or microglia-depleting PLX5622 (PLX) pre-treatment (GEO; GSE160651 dataset). The numbers represent the mean normalized mRNA count for each gene for the indicated experimental group. (**b**) Heatmap depicting differential gene expressions of microglial signature genes across treatment groups using *z*-score transformation of normalized mRNA counts. Relative to the mean value across groups for a given gene, a more orange color indicates a relative increase in gene expression, whereas a bluer color indicates a relative decrease. Significant gene expression changes of >1.5 fold following TBI (*p* < 0.05) are indicated on the left, and the asterisks indicate *p* < 0.05 for the TBI + idebenone (TBI + ideb) to TBI + vehicle (TBI) comparison.

**Figure 4 cells-14-00824-f004:**
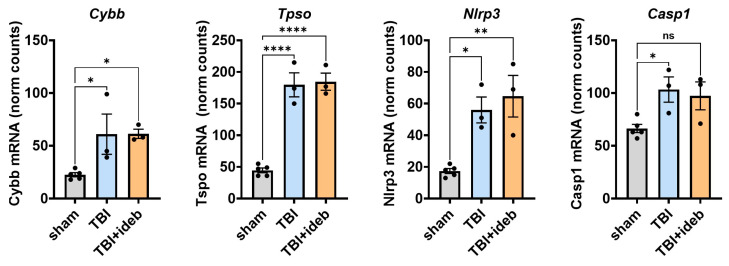
Prototypical TBI neuroinflammation markers are not modified by idebenone treatment. NanoString-quantified gene expression of several pro-inflammatory microglial marker genes upregulated following TBI are shown as normalized (norm) mRNA counts (*n* = 3–5 per group). *p*-values were determined by one-way ANOVA followed by Tukey’s post hoc analysis. * *p* < 0.05, ** *p* < 0.01, **** *p* < 0.0001. Ideb, idebenone; ns, not significant.

**Figure 5 cells-14-00824-f005:**
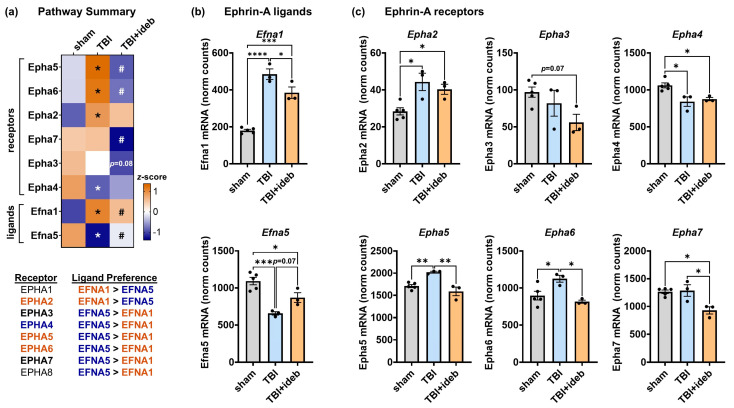
Idebenone mitigates the broad TBI-induced dysregulation of ephrin-A signaling genes. (**a**) Heat map depicting ephrin-A ligand receptor changes across groups, shown as mean *z*-score. Relative to the mean value across groups for a given gene, a more orange color indicates a relative increase in gene expression, whereas a bluer color indicates a relative decrease. An accompanying table lists receptor ligand preferences. The *p*-values in (**a**) were derived from the Rosalind^®^ differential gene expression analysis of the TBI + vehicle (TBI) to TBI + idebenone (TBI + ideb) comparison. * *p* < 0.05 for TBI vs. sham, # *p* < 0.05 for TBI + ideb vs. TBI. (**b**,**c**) NanoString-quantified gene expression of ephrin-A ligands (**b**) and receptors (**c**), shown as normalized (norm) mRNA counts (*n* = 3–5 per group). The *p*-values in (**b**,**c**) were determined by one-way ANOVA followed by Tukey’s post hoc analysis. * *p* < 0.05, ** *p* < 0.01, *** *p* < 0.001, **** *p* < 0.0001.

**Figure 6 cells-14-00824-f006:**
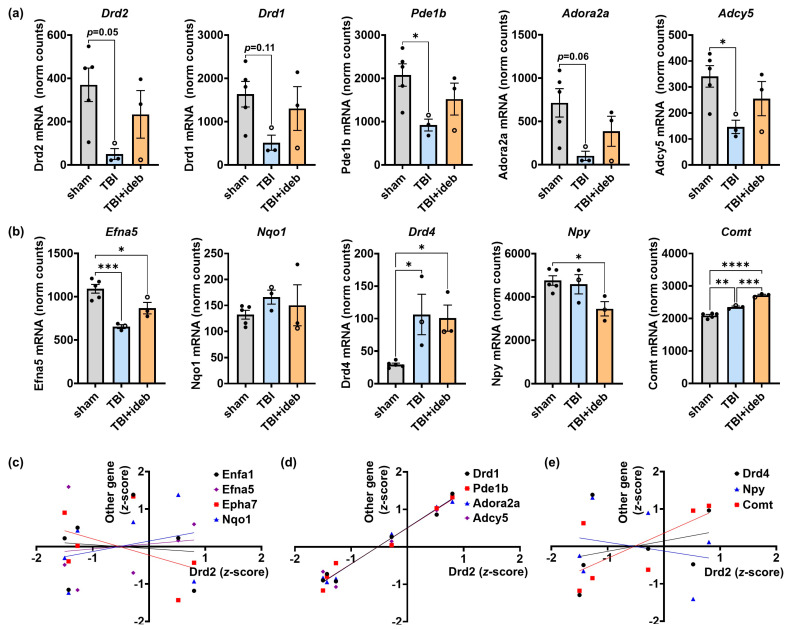
Comparison of gene expression patterns across brain-injured animals reveals a partial responder to idebenone treatment. (**a**) Genes belonging to the GO:BP “behavior” pathway showing a similar pattern of expression to *Drd2* across brain-injured mice, defined as the “*Drd2*-related subgroup”. (**b**) Genes failing to show a *Drd2*-correlated expression pattern. *n* = 3–5 per group for (**a,b**). The *p*-values in (**a**,**b**) were determined by one-way ANOVA followed by Tukey’s post hoc analysis.* *p* < 0.05, ** *p* < 0.01, *** *p* < 0.001, **** *p* < 0.0001. (**c**) Linear regression analysis showing lack of correlated expression between *Nqo1* or representative ephrin-A signaling genes and *Drd2*. (**d**) Linear regression analysis showing significant correlation between the expression of *Drd2* and that of *Drd1*, *Pde1b*, *Adora2*, or *Adcy5*. (**e**) Linear regression analysis showing lack of correlated expression between *Drd2* and *Drd4*, *Npy*, or *Comt*.

**Figure 7 cells-14-00824-f007:**
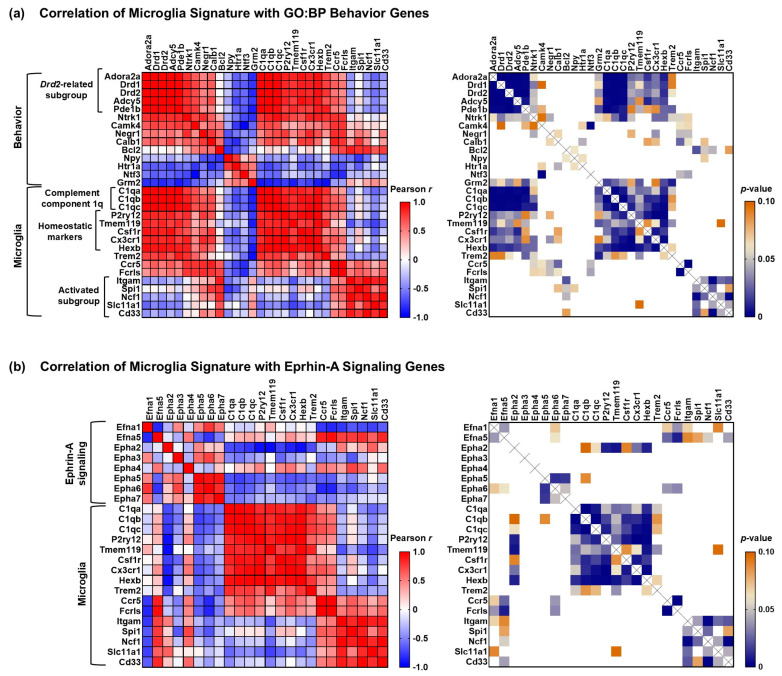
Gene co-expression analysis reveals a relationship between a subgroup of “behavior” genes that include *Drd1* and *Drd2*, complement component 1q (*C1q*) genes, and homeostatic microglial genes following TBI. (**a**) Analysis of gene co-expression between microglial signature and “behavior” genes. (**b**) Analysis of gene co-expression between microglial signature and ephrin-A signaling genes. The heatmaps in the left panels show Pearson’s correlation coefficients (*r*), and the heatmaps in the right panels display *p*-values ≤ 0.1. For the left panels, the redder the color of the box intersecting two genes, the higher the correlation of their gene expression, whereas the bluer the color, the more the two genes were inversely correlated. A white box indicates zero correlation. For the right panels, the bluer the box at the intersection of two genes, the more significant the correlation. Coefficients that were *p* > 0.1 are shown in white.

**Figure 8 cells-14-00824-f008:**
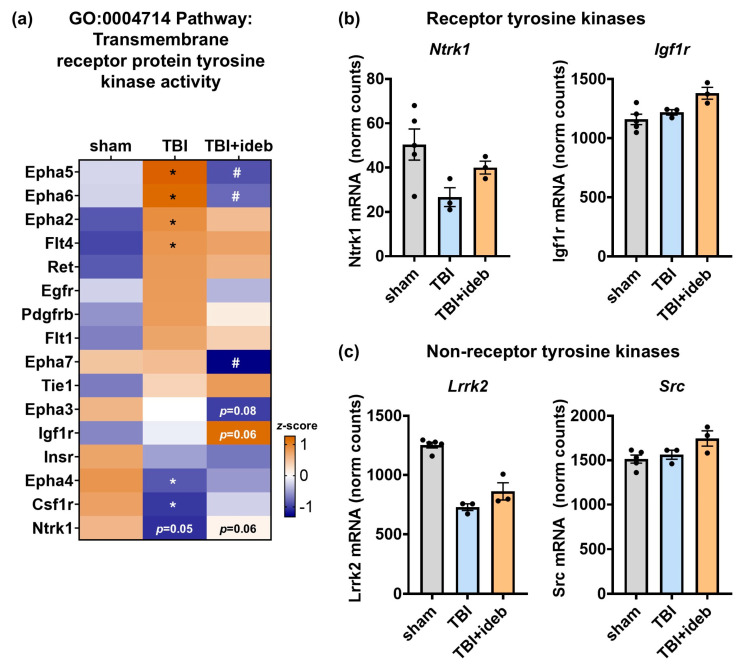
Idebenone modifies the gene expression of several SHC1-interacting tyrosine kinase enzymes following TBI. (**a**) Heat map depicting gene expression changes across Neuropathology Panel members of the transmembrane receptor tyrosine kinase activity gene ontology pathway, with results shown as mean *z*-score. Relative to the mean value across groups for a given gene, a more orange color indicates a relative increase in gene expression, whereas a bluer color indicates a relative decrease. The *p*-values were derived from the Rosalind^®^ differential gene expression analysis of the TBI + vehicle (TBI) to TBI + idebenone (TBI + ideb) comparison. * *p* < 0.05 for TBI vs. sham, # *p* < 0.05 for TBI + ideb vs. TBI. (**b**,**c**) NanoString-quantified gene expression of select receptor tyrosine kinases (**b**) and non-receptor tyrosine kinases (**c**), shown as normalized (norm) mRNA counts (*n* = 3–5 per group).

**Figure 9 cells-14-00824-f009:**
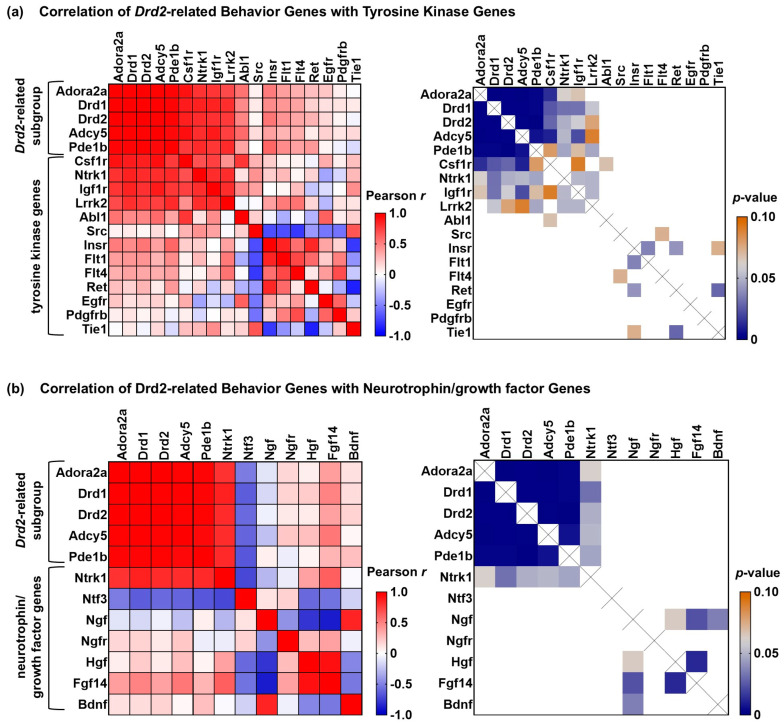
Gene co-expression analysis revealed TRKA-encoding *Ntrk1* and the microglial signature gene *Csf1r* as the tyrosine kinase receptor genes most strongly correlated with *Drd2*-related gene expression following TBI. (**a**) Analysis of gene co-expression between the *Drd2*-related “behavior” gene subgroup and tyrosine kinase genes. (**b**) Analysis of gene co-expression between the *Drd2*-related “behavior” gene subgroup and neurotrophin-/growth-factor-encoding genes. The heatmaps in the left panels show Pearson’s correlation coefficients (*r*), and the heatmaps in the right panels display *p*-values ≤ 0.1. For the left panels, the redder the color of the box intersecting two genes, the higher the correlation of their gene expression, whereas the bluer the color, the more the two genes were inversely correlated. A white box indicates zero correlation. For the right panels, the bluer the box at the intersection of two genes, the more significant the correlation. Coefficients that were *p* > 0.1 are shown in white.

**Figure 10 cells-14-00824-f010:**
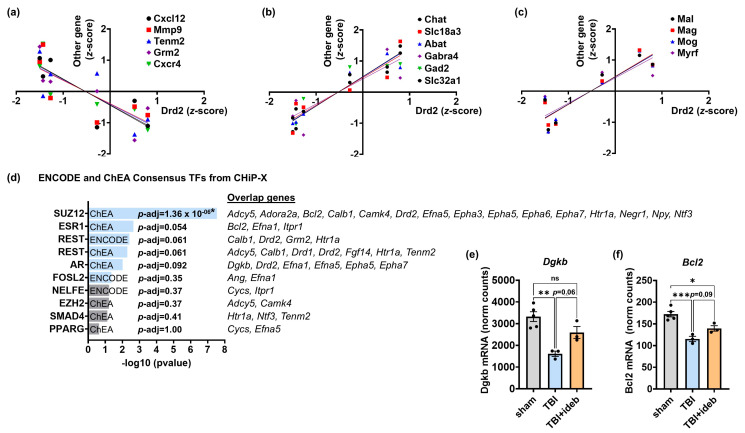
Bioinformatics analyses identified candidate signaling processes and transcription factors for mediating neuropathology and neuroprotection. (**a**) Linear regression analysis showing an inverse correlation between the expression of *Drd2* (dopamine receptor D2) and *Cxcl12* (C-X-C motif chemokine ligand 12), *Mmp9* (matrix metallopeptidase 9), *Tenm2* (teneurin transmembrane protein 2), *Grm2* (glutamate metabotropic receptor 2), or *CxCr4* (C-X-C motif chemokine receptor 4). (**b**) Linear regression analysis showing a positive correlation between the expression of *Drd2* and *Chat* (choline O-acetyltransferase), *Slc18a3* (vesicular acetylcholine transporter), *Abat* (4-aminobutyrate aminotransferase), *Gabra4* (gamma-aminobutyric acid type A receptor subunit alpha4), *Gad2* (glutamate decarboxylase 2), or *Slc32a1* (gamma-aminobutyric acid vesicular transporter, member 1). (**c**) Linear regression analysis showing a positive correlation between the expression of *Drd2* and *Mal* (mal, T cell differentiation protein), *Mag* (myelin-associated glycoprotein), *Mog* (myelin oligodendrocyte glycoprotein), or *Myrf* (myelin regulatory factor). (**d**) The top 10 ENCODE and ChEA consensus transcription factors (TFs) from CHiP-X. The light blue bars indicate *p* < 0.05. False-discovery-corrected adjusted *p*-values (*p*-adj) are shown next to each result, along with the genes targeted by each transcription factor that overlapped with the list affected by idebenone treatment. Note that although the Enrichr output defaults to human genes names, the mouse gene names are shown for consistency. (**e**,**f**) NanoString-quantified gene expression of *Dgkb* (diacylglycerol kinase beta), (**e**) and *Bcl2* (B cell leukemia/lymphoma 2), (**f**), shown as normalized (norm) mRNA counts (*n* = 3–5 per group). The *p*-values in e and f were determined by one-way ANOVA followed by Tukey’s post hoc analysis. ns, not significant. * *p* < 0.05, ** *p* < 0.01, *** *p* < 0.001.

**Table 1 cells-14-00824-t001:** List of genes significantly modified by idebenone treatment after TBI compared with TBI + vehicle treatment (*p* < 0.05). The genes shown in bold encode proteins reported to interact with the idebenone-binding protein SHC1 or with SHC1-interacting receptors. The underlined genes are preferentially expressed by microglia within the healthy mouse brain [18].

Name	Description	Fold Change	*p*-Value
Drd2	dopamine receptor D2	6.19	0.040
Cd4	CD4 antigen	5.30	0.011
Slc18a3	solute carrier family 18 (vesicular monoamine), member 3	3.94	0.037
Drd1	dopamine receptor D1	2.58	0.049
Ang	angiogenin, ribonuclease, RNase A family, 5	1.94	0.044
Dgkb	diacylglycerol kinase, beta	1.61	0.002
Tlr2	toll-like receptor 2	1.45	0.027
Ccr5	chemokine (C-C motif) receptor 5	1.37	0.038
**Efna5**	ephrin A5	1.33	0.011
Calb1	calbindin 1	1.31	0.029
**Hgf**	hepatocyte growth factor	1.31	0.043
Itpr1	inositol 1,4,5-trisphosphate receptor 1	1.31	0.013
Stab1	stabilin 1	1.30	0.036
Hpgds	hematopoietic prostaglandin D synthase	1.28	0.009
Camk4	calcium/calmodulin-dependent protein kinase IV	1.25	0.027
**Fgf14**	fibroblast growth factor 14	1.25	0.048
Negr1	neuronal growth regulator 1	1.24	0.009
Bcl2	B cell leukemia/lymphoma 2	1.23	0.025
Fcrls	Fc receptor-like S, scavenger receptor	1.21	0.001
Gusb	glucuronidase, beta	1.20	0.009
C1qb	complement component 1, q subcomponent, beta polypeptide	1.18	0.043
**Lrrk2**	leucine-rich repeat kinase 2	1.18	0.047
** Grn **	granulin	1.18	0.011
Gucy1b3	guanylate cyclase 1, soluble, beta 3	1.17	0.041
Amph	amphiphysin	1.16	0.013
Comt	catechol-O-methyltransferase	1.16	0.000
Cntn4	contactin 4	1.15	0.015
Ap1s1	adaptor protein complex AP-1, sigma 1	1.14	0.028
Nptn	neuroplastin	1.14	0.029
Tnr	tenascin R	1.12	0.033
U2af2	U2 small nuclear ribonucleoprotein auxiliary factor (U2AF) 2	1.12	0.008
Gsr	glutathione reductase	1.09	0.011
Cycs	cytochrome c, somatic	−1.27	0.030
**Efna1**	ephrin A1	−1.28	0.022
**Epha5**	Eph receptor A5	−1.28	0.001
Tenm2	teneurin transmembrane protein 2	−1.33	0.003
Npy	neuropeptide Y	−1.33	0.020
Grm2	glutamate receptor, metabotropic 2	−1.38	0.038
Epha6	Eph receptor A6	−1.39	0.007
Epha7	Eph receptor A7	−1.39	0.005
Htr1a	5-hydroxytryptamine (serotonin) receptor 1A	−1.85	0.002
**Ntf3**	neurotrophin 3	−2.90	0.035

**Table 2 cells-14-00824-t002:** NanoString gene set analysis for the TBI + vehicle vs. TBI + idebenone comparison. Five genes changing by >1.5-fold with a *p*-value between 0.05 and 0.1 (*Adcy5*, *Adora2a*, *Epha3*, *Ntrk1*, and *Pde1b*, see Table 3) are included and indicated by gray font.

Term	Significance Score	Genes Altered
Tissue Integrity	1.8404	Cd4, Efna1, Efna5, Epha3, Epha5, Epha6, Epha7, Negr1
Transmitter Synthesis and Storage	1.4634	Comt, Drd1, Drd2, Htra1a, Pde1b, Slc18A3
Axon and Dendrite Structure	1.4395	Ap1s1, Calb1, Cntn4, Comt, Drd1, Drd2, Efna5, Epha7, Grm2, Lrrk2, Nptn, Tenm2
Transmitter Release	1.3826	Adcy5, Adora2a, Bcl-2, Camk4, Comt, Drd1, Drd2, Grm2, Htra1a, Itpr1, Pde1b
Neural Connectivity	1.3675	Amph, Ap1s1, Calb1, Comt, Drd1, Drd2, Epha7, Grm2, Itpr1, Lrrk2, Nptn, Slc18a3, Tenm2
Trophic Factors	1.3475	Bcl-2, Camk4, Ntf3, Ntrk1
Transmitter Response and Reuptake	1.3387	Adcy5, Adora2a, Camk4, Drd1, Drd2, Grm2, Gucy1b3, Itpr1, Nptn, Npy
Growth Factor Signaling	1.2998	Bcl2, Drd2, Efna1, Efna5, Epha3, Epha5, Epha6, Fgf14, Hgf, Ntf3, Ntrk1
Vesicle Trafficking	1.2817	Amph, Calb1, Cntn4 Drd1, Drd2, Fgf14, Grm2, Lrrk2, Nptn, Npy, Ntf3, Ntrk1, Slc18a3, Tnr
Activated Microglia	1.2704	Bcl-2, C1qb, Ccr5, Fcrls, Grn, Gusb, Tlr2

**Table 3 cells-14-00824-t003:** Genes altered by >1.5-fold by idebenone following TBI with *p*-values between 0.05 and 0.1.

Name	Description	Fold Change	*p*-Value
Adora2a	adenosine A2a receptor	4.23	0.057
Ntrk1	neurotrophic tyrosine kinase, receptor, type 1	2.12	0.062
Adcy5	adenylate cyclase 5	1.82	0.057
Pde1b	phosphodiesterase 1B, Ca^2+^-calmodulin dependent	1.66	0.068
Epha3	Eph receptor A3	−1.71	0.078

## Data Availability

The original contributions presented in this study are included in the article/Appendix A. Further inquiries can be directed to the corresponding author.

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
