# Peer review of "Idebenone Mitigates Traumatic-Brain-Injury-Triggered Gene Expression Changes to Ephrin-A and Dopamine Signaling Pathways While Increasing Microglial Genes"

_cells, 2025, doi:10.3390/cells14110824_

Round 1

Reviewer 1 Report

Comments and Suggestions for Authors

Hyehyun Hwang et al. present a study investigating the effects of idebenone on the early microglial response to TBI, with a focus on transcriptomic alterations. New molecular understanding is provided by the discovery that idebenone targets the dopamine and ephrin-A signaling pathways. The findings' interpretive framework is strengthened by the authors multi-tiered analytical strategy, which includes GO, KEGG pathway mapping, gene co-expression analysis, and transcription factor enrichment.

However, following concerns should be addressed.

  1. Based just on gene expression patterns, it may be concluded that idebenone encourages the growth of healthy microglia. This conclusion is still dubious in the absence of complementary validation, such as immunophenotyping or cell-type-specific labeling.
  2. Idebenone's effects on neuroinflammation and recovery cannot be evaluated for durability or functional significance because the study is restricted to a single acute time point (24 hours post-TBI). To ascertain if these molecular alterations result in long-lasting neuroprotection or better behavioral outcomes (e.g., immunostaining, behavior tests), longitudinal studies would be helpful.

Reviewer 2 Report

Comments and Suggestions for Authors

The article is well thought out and written, but some older references from 1994, for example, could be replaced with more recent references.

Reviewer 3 Report

Comments and Suggestions for Authors

This text deals with the relevant issue of microgial activation and its possible participation in neurodegenerative disorders; however, some issues need to be clarified:

  1. The title (very long) describes a rather unexpected finding of glial activation, more than inhibition, which was the goal of this test. A title should describe this experiment, rather than early findings which are clearly described in the text.
  2. The potential activation of microglia as the factor for late disorders in a hypothesis, this feature should be explained in the text from this initial therapeutic study.
  3. Figures are too many (10) as well as supplemental (several tables) which are difficult to real and comprehend, in my opinion illustrations should be minimized and briefly explained in the text, the experiments and results of this incipient trial made in experimental mice could incite further clinical testing (P21,L658-664) before defining the potential use of Idebenone as adjuvant therapy of brain trauma.
  4. As a potential limitation of this study a brief discussion on cost-benefit could be included on P20,L640-654.

Round 2

Reviewer 1 Report

Comments and Suggestions for Authors

All raised questions have been carefully addressed by the author, and the responses have been added in the appropriate locations.